# Reliability and Validity of a Smartphone Device and Clinical Tools for Thoracic Spine Mobility Assessments

**DOI:** 10.3390/s23177622

**Published:** 2023-09-02

**Authors:** Gabriela Bella van Baalen, Benedicte Vanwanseele, Ranel Rachel Venter

**Affiliations:** 1Department of Exercise, Sport and Lifestyle Medicine, Stellenbosch University, Stellenbosch 7600, South Africa; gtidbury@sun.ac.za (G.B.v.B.); rev@sun.ac.za (R.R.V.); 2Department of Movement Sciences, KU Leuven, 3001 Leuven, Belgium

**Keywords:** biomechanics, bubble inclinometer, goniometer, mobile application, range of motion (ROM), reliability, smartphone, thoracic spine, validity

## Abstract

Quantifying thoracic spine mobility with reliable and valid tools is a challenge for clinicians in practice. The aim of this study is to determine the reliability and validity of a smartphone device, bubble inclinometer and universal goniometer to quantify the static kyphotic curve and active range of motion of the thoracic spine. A total of 17 participants (mean age = 23.7 ± 2.3 years) underwent repeated measurements with three raters, on three separate days performing the lumbar-locked trunk rotation, standing full extension, standing full flexion, standing relaxed curve and seated trunk rotation assessments. Mostly “Good” to “Excellent” intra-rater (ICC ranging from 0.624 to 0.981) and inter-reliability (ICC ranging from 0.671 to 0.968) was achieved with the smartphone and clinical tools. “Excellent” validity (ICC ranging from 0.903 to 0.947) with the smartphone was achieved during lumbar-locked trunk rotation and standing relaxed curve assessment. “Good” validity (ICC ranging from 0.836 to 0.867) of the smartphone was achieved during the seated trunk rotation. The Samsung s9+ smartphone with the iSetSquare mobile application is a reliable and valid tool to use for clinical assessments assessing thoracic spine mobility.

## 1. Introduction

Optimal thoracic spine mobility is important for functional movement of the body. The thoracic spine largely contributes to spinal rotation, flexion and extension in sporting movements such as swimming, surfing, standup paddle boarding, boxing, rowing, golf, rugby, netball, hockey, soccer and overhead throwing sports [1,2,3].

In clinical practice it is important to quantify the range of motion to aid clinical decision making and track progressions of interventions applied by medical practitioners such as doctors, physiotherapists, occupational therapists and biokineticists [4,5]. Quantifying the range of motion proves to be a challenge because of its complex structure [2].

There is limited research reporting reliability and validity of the tools used to quantify these movements because most studies focus on quantifying the passive thoracic kyphotic curvature [6,7,8,9] and few studies focus on active thoracic spine rotation [1,2,3,10]. Tools such as the universal goniometer, bubble inclinometer and smartphones are often used to quantify range of motion [11]. When it comes to determining joint range of motion, the universal goniometer [5] is known to be the industry standard but X-ray imaging is the gold standard for determining the angle of the curvatures of the spine [12].

Inconsistencies within research methodologies such as the tools used, assessments and population of interest makes it hard to find the golden thread between research and clinical application of which tools and assessments are valid, reliable and have clinical significance in determining thoracic spine mobility. However, Johnson and Grindstaff [13] published a guideline for thoracic spine rotation assessments in 2010 with the aim of providing clear instructions on how to conduct these assessments.

With technology advancing at a rapid rate, many people own a smartphone [14] giving them access to mobile applications such as inclinometers that are cost effective, non-invasive and easy to use [5,15,16]. Modern smartphones are built with gyroscopes, accelerometers and digital magnetometer sensors, turning the device into inertial motion units (IMUs) [17]. Therefore, mobile applications can process the data from these sensors to allow the smartphone to perform like an inclinometer. Most smartphones are small enough to fit into a pocket, occupying a small space (roughly 7 cm wide and 14.7 cm high) [18] and can perform measurements in muti-planes. Typically, with the traditional methods multiple tools would be needed to measure the sagittal, frontal and transverse planes in clinical practice. Previous studies have mainly determined the reliability and validity quantifying the range of motion of the ankle, knee, hip, spine, elbow and wrist [5,9,11,15,19].

Currently there are no recent reliability and validity studies assessing thoracic spine mobility with recently released smartphones. This study will be one of the first studies reporting the repeatability and accuracy of the smartphone to measure thoracic spine mobility. The aim of this study is to determine the reliability and validity of a smartphone device, bubble inclinometer and universal goniometer for thoracic spine mobility assessments. The two objectives of this study are to determine: (i) the intra-rater and inter-rater reliability of a smartphone device and common or preferred clinical tools used to measure thoracic spine mobility in asymptomatic adults; (ii) the validity of a smartphone with commonly used clinical tools that measure thoracic spine mobility in asymptomatic adults. We hypothesized that the smartphone device will provide equivalent ICC categorical outcomes for reliability as the commonly used clinical tools. In addition, we hypothesized that the smartphone device will produce five valid results with minimal bias.

## 2. Materials and Methods

### 2.1. Ethics

The study was conducted in accordance with the Declaration of Helsinki and approved by the Research Ethics Committee: Human Research Ethics Committee (HUMANIORA) at Stellenbosch University, South Africa (project number: S19/08/164 PhD, 4 June 2020). Written informed consent was obtained from all subjects involved in the study.

### 2.2. Study Design

This study follows an observational study design.

### 2.3. Participants

A total of 17 participants (13 females and 4 males) aged 20 to 28 years old (mean ± standard deviation [SD]; age = 23.7 ± 2.3 years; height = 1.69 ± 0.1 m; body weight = 63.8 ± 8.9 kg; body mass index [BMI] = 22.3 ± 1.6 kg/m^2^) volunteered for this study. Participants were recruited by social media platforms, posters and flyers placed on notice boards throughout Stellenbosch University campus and Stellenbosch, Western Cape, South Africa. Participants interested in partaking in the study signed up with their personal details via Microsoft Forms (Microsoft 365 (Office), Microsoft Corporation, Redmond, WA, USA). The researcher contacted each participant via email and sent them an informed consent form, Physical Activity Readiness Questionnaire (PAR-Q) [20] and a medical health screening form.

Participants were included in the study if they completed the above-mentioned forms, if their age was between 18–35 years old to eliminate age-related spine degeneration [21], were asymptomatic and presented with no injuries within three months prior to the testing date. Participants were excluded if their standing relaxed thoracic curve was equal or greater than 40° (hyperkyphosis) [22], had a history of spinal disorders such as: scoliosis (diagnosed by a medical professional), Scheuermann’s disease, Ankylosing spondylitis, spinal fractures, osteoarthritis, osteoporosis, disc herniation or received spinal surgery. Any participant that received medical treatment for back or neck pain six months prior to testing was excluded. In addition, participants were excluded if their body mass index (BMI) was greater than 25 kg/m^2^ according to the World Health Organization (WHO) [23] because the excess weight can influence the ability to locate the spinal processes.

### 2.4. Clinical Tools (Equipment)

The reliability and validity of the following tools: Samsung s9+ smartphone (SM-G965F, Samsung, Yateley, UK) with the mobile application: iSetSquare (Version 1.2., Plaincode, Stephanskirchen, Germany), bubble inclinometer (SenseAid, SILV, Madison, WI, USA) and universal goniometer (HiTech Therapy, Randburg, South Africa) were determined during five clinical assessments for thoracic spine mobility. Refer to Table 1 for a summary of tools and clinical assessments performed.

### 2.5. Observers (Raters)

Three exercise rehabilitation specialists with clinical experience ranging from two to twenty years took part in the study to determine the intra-rater and inter-rater reliability and validity of the clinical tools. Each rater received a testing protocol to familiarize themselves with the testing procedures and a one-on-one practical demonstration of the clinical assessments and tools.

### 2.6. Procedures

To determine if the volunteers fulfilled the inclusion criteria, weight, height and standing relaxed curve (refer to Section 2.6.1 for an explanation of this assessment) were measured during the first visit. All participants that fulfilled the inclusion criteria of this study performed the clinical assessments summarized in Table 1 in a randomized order with each rater. Each rater recorded three trials per clinical assessment and tool. The testing order of the clinical assessments and usage of tool was randomized. The testing procedure was repeated on three separate days.

Before each rater commenced with the testing procedure, they would request each participant to stand comfortably with their feet below their hips and instruct them to look at a target that was placed on the wall at eye level with their arms relaxed next to their sides. Each rater palpated the vertebral spinal processes of the participants spine and marked spinal processes of C7 to L4 with a white board marker. Position one (P1) was located at T1 spinal process and position two (P2) was at T12 spinal process. P1 and P2 served as measurement sites for the clinical assessments. The dots on the skin were easily wiped off after testing with alcohol wet wipes. No warm-up was given because there was minimal risk for injury with each movement and to eliminate the effect fatigue may have on the participant by completing five clinical assessments with three raters on the same day.

#### 2.6.1. Lumbar-Locked Trunk Rotation

Participants were requested to sit back onto their heels, knees slightly apart, with their forearms parallel with the surface of the plinth (quadruped position). Elbows were placed just above their knees under their shoulders. The rater would help position the participant’s head into neutral position and instruct the participant to maintain this posture. If rotation to the right was being performed, the rater would place the tool on P1 (set the tool to zero), instruct the participant to place their right hand onto the left shoulder and turn their torso to the right side as far as possible (see Figure 1). They would hold this position until the measurement was recorded. The same procedure would follow with rotation to the left, except the left hand would be placed on the right shoulder.

#### 2.6.2. Standing Full Extension

The participant was requested to stand relaxed with their feet hip-width apart and look at a target on the wall while the rater placed the devices on P1 and P2. The rater instructed the participant to place their hands on their hips, push their chest forward and lean back as far as possible without falling over (see Figure 2). They would hold this position until the rater recorded the measurements at P1 and P2.

#### 2.6.3. Standing Full Flexion

The participant was requested to stand relaxed with their feet hip-width apart and look at the target on the wall while the rater placed the devices on P1 and P2. The rater requested the participant to bend forward and relax completely. The rater would apply some pressure at the superior aspect of their back to ensure that they have rounded their back fully (see Figure 3). A measurement at P1 and P2 was recorded.

#### 2.6.4. Standing Relaxed Curve

The participant was requested to stand relaxed with their feet hip-width apart, arms relaxed next to their sides and look at the target on the wall. The rater would set the tool to zero as they placed it on P1 and then moved the device to P2 to record the measurement.

#### 2.6.5. Seated Trunk Rotation

The participant was instructed to sit on the edge of the plinth with their knees almost touching the edge of the plinth with a foam roller between their knees. They were instructed to hold a dowel across their chest with their arms crossed, dowel level and sit upright. The rater set the tool to zero as they placed the tool on P1. If rotation to the right was performed, the rater requested the participant to squeeze the foam roller and turn as far as possible to the right side. They would hold this position until the rater recorded the measurement (see Figure 4). The same procedure would be followed for rotation to the left, except they would be instructed to rotate to the left side.

### 2.7. Statistical Analysis

All raw data were stored in Microsoft Excel (Microsoft 365 (Office), Microsoft Corporation, Redmond, WA, USA) and imported into IBM SPSS (version 28, SPSS Inc., Chicago, IL, USA) software for statistical analysis. The alpha value was set at 0.05 for all analyses. The Shapiro-Wilk test and Q-Q plots were performed with all variables to determine the distributions thereof. Most variables were normally distributed, and sensitivity tests were performed on the non-normally distributed variables by performing log transformation to determine whether the outcomes of these variables changed after repeating the analysis on the log-transformed data. All log-transformed variables were not normally distributed after the log 10 transformation except one variable. However, after repeating the analysis on these variables, there were no significant changes with the outcomes of these variables, even the one variable that was normally distributed post-log transformation. Therefore, all statistical analysis was performed on all normally and non-normally distributed raw data.

#### 2.7.1. Reliability

Intra-rater and inter-rater reliability was determined by Interclass Correlation Coefficient (ICC) and 95% confidence intervals based on Two-way random model, mean-rating (k = 3) and absolute agreement [24]. The interpretation of the ICC values was considered poor if ICC < 0.50, moderate if 0.50 ≤ ICC < 0.75, good if 0.75 ≤ ICC < 0.90 or excellent if ICC ≥ 0.90 [24]. In addition, Standard Error of Measurement (SEM = SD × $\sqrt{1\;-\;\mathrm{ICC}}$) and Minimal Detectable Change (MDC_95_) were calculated for intra-rater and inter-rater reliability [25].

#### 2.7.2. Validity

Validity between preferred tools and smartphone was determined by Interclass Correlation Coefficient (ICC), Two-way random model, absolute agreement with 95% confidence intervals. The interpretation of the ICC values was considered poor if ICC < 0.50, moderate if 0.50 ≤ ICC < 0.75, good if 0.75 ≤ ICC < 0.90 or excellent if ICC ≥ 0.90 [24].

Bland–Altman plots were used to determine the level of agreement [26] and linear regression analysis was used to determine the proportional bias.

## 3. Results

### 3.1. Reliability

#### 3.1.1. Intra-Rater Reliability

Table 2 represents the intra-rater reliability of three raters performing thoracic spine mobility assessments over separate days.

##### Lumbar Locked Trunk Rotation

For rotation to the left, R1 and R2 achieved “Good” reliability with the preferred tool and smartphone. However, R3 achieved “Excellent” reliability with the preferred tool and smartphone. When looking at rotation to the right, all raters achieved “Good” reliability with the preferred tool and smartphone with one exception of R1 achieving “Moderate” reliability with the smartphone. Refer to Table 2 for ICC, 95% confidence intervals (CI) and SEM.

##### Standing Full Extension

Both R1 and R2 achieved “Excellent” reliability and R3 achieved “Good” reliability at both measurement sites.

##### Standing Full Flexion

At P1 measurement site R1 achieved “Excellent” reliability and both R2 and R3 achieved “Good” reliability. All three raters achieved “Excellent” reliability at P2 measurement site.

##### Standing Relaxed Curve

R1 achieved “Excellent” reliability with the preferred tool and smartphone. R2 achieved “Moderate” reliability with the preferred tool and smartphone. R3 achieved “Good” reliability with preferred tool and smartphone.

##### Seated Trunk Rotation

For rotation to the left, all raters achieved “Excellent” reliability with the preferred tool and smartphone except R3 achieved “Good” reliability with the preferred tool. For rotation to the right, both R1 and R3 achieved “Excellent” reliability and R2 achieved “Good” reliability with the smartphone. R1 and R2 achieved “Excellent” reliability and R3 “Moderate” reliability with the preferred tool.

#### 3.1.2. Inter-Rater Reliability

Table 3 represents the inter-rater reliability between three raters over three separate days.

##### Lumbar-Locked Trunk Rotation

“Excellent” reliability was achieved with rotation to the left and “Good” reliability was achieved with rotation to the right for the preferred tool and smartphone. Refer to Table 3 for ICC, 95% confidence intervals (CI) and SEM.

##### Standing Full Extension and Standing Full Flexion

“Excellent” reliability for both measurement sites was achieved for the standing full extension and standing full flexion assessment.

##### Standing Relaxed Curve

“Good” reliability with the preferred tool was achieved and “Moderate” reliability was achieved with the smartphone.

##### Seated Trunk Rotation

“Excellent” reliability was achieved with rotation to the left and right with the smartphone. “Excellent” reliability was achieved with rotation to the left and “Good” reliability was achieved with rotation to the right with the preferred tool.

### 3.2. Validity

Refer for Table 4 for descriptive data of validity results and Bland-Altman plot residual means. Statistically significant *p* values (*p* < 0.001) for all tests were recorded. In addition, refer to Table 5 for the ICC’s values and Figure 5 for Bland-Altman plots with linear regression lines.

#### 3.2.1. Lumbar-Locked Trunk Rotation

“Excellent” validity between the smartphone and preferred tool with good agreement was achieved with rotation to the left and right and no significant proportional bias present (left rotation: *p* = 0.76; right rotation: *p* = 0.80).

#### 3.2.2. Standing Relaxed Curve

“Excellent” validity between the smartphone and preferred tool with good agreement was achieved and no significant proportional bias present (*p* = 0.27).

#### 3.2.3. Seated Trunk Rotation

“Good” validity between the smartphone and preferred tool was achieved with moderate agreement with rotation to the left and good agreement with rotation to the right. In addition, significant proportional bias present with rotation to the right (*p* < 0.001) and to the left (*p* = 0.01).

## 4. Discussion

This study investigated the reliability and validity of a smartphone and preferred tools used to assess the thoracic spine mobility with five assessments namely: lumbar-locked trunk rotation, standing full flexion, standing full extension, standing relaxed curve and seated trunk rotation. The intra-rater and inter-rater reliability and validity will be discussed as per assessment.

### 4.1. Lumbar-Locked Trunk Rotation

#### 4.1.1. Intra-Rater Reliability

Lumbar-locked trunk rotation assesses thoracic spine rotation in a four-point position with the participant sitting back onto their heels. When the hips and lower back are in maximal flexion, it is believed that it helps to stabilize the pelvic region to prevent compensatory movements during the assessment [1].

Each rater consistently achieved similar intra-rater reliability outcomes between the smartphone and the bubble inclinometer when comparing the results to themselves. However, differences between the raters and side of rotation were noticeable. The consistency of each rater achieving similar results with both tools is an indication that the variability found between the raters could be because of not having enough experience conducting the lumbar-locked trunk rotation assessment. Secondly, this was the first time the raters conducted this assessment themselves. Although the clinical experience between each rater differs between one another, each rater managed to obtain comparable results between the smartphone and preferred tool. In general, all three raters had slightly lower intra-rater reliability outcomes with rotation to the right. The differences between the outcomes with rotation to the right and the left could have occurred from the participants’ dominance to rotate to one side more. However, this would require further investigation to determine which side is the dominant side of the participants and is a limitation of this study. Overall, the smartphone achieved better agreement than the bubble inclinometer with rotation to the right.

The difference between right and left rotation with the bubble inclinometer also occurred in the study of Johnson et al. [1]. The results of a recent study published by Feijen et al. [10] achieved “Excellent” intra-rater reliability with good agreement for the bubble inclinometer. To date there are no studies reporting the intra-rater reliability of the smartphones and precaution should be taken when comparing the results of the smartphone to studies performed with the bubble inclinometer. In addition, one needs to note that Feijen et al. [10] did not randomize the order of their raters. Despite achieving comparable results to Johnson et al. [1] and Feijen et al. [10], precaution should be taken when interpreting the result of this current study because the sample size of this study was smaller, and wider 95% CI intervals were achieved with rotation to the right. Not only did the smartphone produce “Good” to “Excellent” intra-rater reliability indicating that it can repeatedly reproduce measurements taken by same clinician, but it also produced clinically acceptable SEM for rotation to the right and left.

#### 4.1.2. Inter-Rater Reliability

A difference between the rotation to the right and left inter-rater reliability was noticeable. Larger 95% CI was achieved with rotation to the right indicating large variability within the measurements. Despite the differences with the 95% CI between rotation to the right and left with both tools, similar measurement errors and minimal detectable change were achieved to each side. The results of the bubble inclinometer with this current study agreed with the results of Johnson et al. [1] that achieved similar ICC outcomes for rotation to both sides. Feijen et al. [10] investigated the inter-rater reliability of a bubble inclinometer between two raters on the same day and achieved slightly higher SEM with larger MDC. Precaution should be taken when interpreting the Feijen et al. [10] results because they did not randomize the testing order and inter-rater reliability was determined by two raters on the same day.

To date there has not been a study reporting the reliability of a smartphone during the lumbar-locked trunk rotation assessment. Therefore, precaution should be taken when comparing the results of the smartphone with the bubble inclinometer. One needs to acknowledge that there was a greater 95% CI for rotation to the right, and this difference might have been resulted from the intra-rater variability found within each rater’s measurements. The small sample size might have also played a role in negatively influencing the 95% CI. Both the smartphone and bubble inclinometer produced “Good” to “Excellent” inter-rater reliability that produced clinically acceptable SEM ranging between 2.3° and 2.8° with MDC of 6.4° to 7.7°. This indicates that a measurement difference of 2.3° to 2.8° may occur with the smartphone and bubble inclinometer, but a minimum change of pre- to post-measurements greater than 6.4° to 7.7° will indicate a significant change.

#### 4.1.3. Validity

“Excellent” validity with the smartphone was achieved with no significant proportional bias present. Bucke et al. [3] achieved higher mean differences with the smartphone of 4.94° comparing smartphone and digital inclinometer to measurements obtained with an ultrasound machine. The smartphone in this current study produced better validity results with good agreement than Bucke et al. [3]. One needs to note that Bucke et al. [3] compared the smartphone to the results of an ultrasound machine and in this current study, it was compared to the bubble inclinometer (industry standard). The smartphone tended to overestimate measurements compared to the bubble inclinometer, but the mean differences were clinically acceptable. This indicates that the smartphone will accurately measure the thoracic spine rotation ROM during the Lumbar-locked trunk rotation assessment with a consistent small measurement difference compared to the bubble inclinometer. Therefore, it is advised that a clinician should use the smartphone throughout their measurements and not use it interchangeably with the bubble inclinometer.

### 4.2. Standing Full Flexion and Extension

#### 4.2.1. Intra-Rater Reliability

Standing full flexion and extension results were only measured with the bubble inclinometer. The reason for this is that it is difficult to measure two measurement sites simultaneously with two smartphones and this could influence the accuracy of the results. Therefore, only intra-rater and inter-rater reliability was determined with the bubble inclinometer for standing full flexion and extension assessment.

Mostly “Excellent” intra-rater reliability with good agreement was achieved for standing full flexion and extension assessments. To the knowledge of the researcher, no research to date has investigated the intra-rater and inter-rater reliability of a bubble inclinometer to measure the isolated flexion and extension range of motion of the thoracic spine. However, a few reliability and validity studies measured lumbar flexion and extension of the spine with the smartphone and bubble inclinometer [27,28,29]. These studies’ methodologies vary with regards to their measurement sites, but the movement carried out is similar to the movement carried out with the standing full flexion and extension assessments in the current study. Kolber et al. [29] achieved similar intra-rater reliability with the bubble inclinometer during the total thoracolumbar-pelvic flexion and the thoracolumbar-pelvic extension assessments. Pourahmadi et al. [28] also achieved similar intra-rater reliability results with a gravity-based inclinometer as to Kolber et al. [29]. These results were like the results achieved by one of the raters within this current study. Precautions should be taken when comparing the results of this current study to the above-mentioned studies because they measured the lumbar flexion and extension and did not isolate the thoracic spinal flexion and extension. Therefore, the bubble inclinometer produced mostly “Excellent” intra-rater reliability with good agreement. This indicates that the bubble inclinometer can be used repeatedly by the same clinician to carry out multiple measurements.

#### 4.2.2. Inter-Rater Reliability

“Excellent” inter-rater reliability with good agreement was achieved with the bubble inclinometer for standing full flexion and extension assessments. This agreed with Kolber et al. [29] that achieved good inter-rater reliability with thoracolumbar-pelvic flexion and the thoracolumbar-pelvic extension assessment. In addition, this current study’s SEM and MDC were like the results published by Pourahmadi et al. [28]. As mentioned above, the results should be compared with caution because these studies investigated lumbar flexion and extension, and not thoracic spine flexion and extension. In summary, the bubble inclinometer achieved “Excellent” inter-rater reliability with good agreement for the standing full flexion and extension assessments. This finding indicates that raters or clinicians can interchangeably take measurements between each other, as well as on separate days.

### 4.3. Standing Relaxed Curve

#### 4.3.1. Intra-Rater Reliability

The standing relaxed curve assessment determines the static thoracic curvature of the spine while standing in a relaxed posture. A curvature greater than 40 degrees classifies an individual with hyperkyphosis [22] and this excessive curve can increase the risk for shoulder and neck-related injuries [9].

All three raters consistently received the same intra-rater reliability interpretation for the smartphone and bubble inclinometer when comparing the results to themselves. One should note that there was a difference between all three raters’ scores. The differences found between the raters may be because of inexperience to perform the assessment, as the standing relaxed curve was a new assessment introduced to the raters. R1 achieved the best result of “Excellent” intra-rater reliability with good agreement and small SEM with the smartphone and bubble inclinometer, which indicates that it is possible to obtain “Excellent” intra-rater reliability. Two of the three raters produced similar intra-rater reliability results similar to Shari and Hesar [9], with one rater over three days with a smartphone (Goniometer-Pro mobile application). One major difference to highlight from the above-mentioned study is that the researchers requested the individuals to keep their shoulders at 90 degrees flexion while taking the measurements, while one rater marked the measurement sites and did not report on the SEM. In addition, they also achieved similar intra-rater reliability results with the bubble inclinometer as Lewis and Valentine [8] with asymptomatic and symptomatic shoulder participants. Despite the differences between the three raters, each one consistently obtained similar results between the smartphone and bubble inclinometer, indicating that the differences between the raters is most likely due to the inexperience to conduct the assessment. We can assume that the smartphone and bubble inclinometer can consistently record the measurements well if the same rater reproduces the measurements.

#### 4.3.2. Inter-Rater Reliability

“Good” inter-rater reliability with bubble inclinometer was achieved with moderate agreement but the smartphone performed poorly, achieving “Moderate” inter-rater reliability with poor agreement between the three raters. Shari and Hesar [9] achieved better inter-rater reliability with good agreement between three raters on the same day with a smartphone compared to the results found in this study. The one difference to note is that Shari and Hesar [9] calculated the inter-rater reliability of repeated measures on the same day and the current study reports on the inter-rater reliability of measurements repeated over three separate days. Based on the results of this current study, the smartphone is not clinically recommended to be used interchangeably between raters or clinicians for repeated measurements. In other words, one person should always use the smartphone to obtain measurements with measurements repeated over multiple days. The variability of rater 2’s intra-rater reliability could have influenced the inter-rater reliability results of the three raters over three days with the smartphone. As mentioned above, this might have been from a lack of experience to perform the assessment effectively. Although small SEM was achieved with the smartphone, the results should be interpreted with caution as poor inter-rater reliably agreement was found.

#### 4.3.3. Validity

The smartphone achieved “Excellent” validity with good agreement and no significant proportional bias present when compared to the bubble inclinometer. This was similar to the results found by Shari and Hesar [9] although they had a much larger reference range of −9.0° to 12.3° for Bland Altman plot, indicating more variance within their results with the smartphone. The smartphone produced valid results when compared to the bubble inclinometer with good agreement but a mean difference of −2.07° in comparison to the bubble inclinometer. Therefore, the smartphone tends to underestimate the measurement values by −2.07° when compared to the bubble inclinometer. The smartphone is a valid tool for measuring the standing relaxed curve of individuals. Clinically it is recommended that the smartphone and bubble inclinometer should not be used interchangeably. The smartphone has a small tendency to underestimate the actual value, but the smartphone is accurate at measuring the standing relaxed curve.

### 4.4. Seated Trunk Rotation

#### 4.4.1. Intra-Rater Reliability

The seated trunk rotation assessment is often used to determine the range of motion of active thoracic rotation with a universal goniometer or by visual estimations [1]. Mostly “Excellent” intra-rater reliability with good agreement was achieved with the smartphone and universal goniometer for rotation to the right and left. Furness et al. [2] achieved slightly higher intra-rater reliability in their study for the smartphone and universal goniometer with two raters. However, precaution should be taken when interpreting their results as they only looked at the raters’ repeated scores for one day and not over multiple days. In addition, the SEM and MDC of this current study were also in agreement with Furness et al. [2]. In addition, R1 and R2 achieved similar intra-rater reliability results with the universal goniometer as Johnson et al. [1]. Most of the raters achieved slightly better intra-rater reliability results with the smartphone compared to the universal goniometer. This could be because it is easier to calibrate the phone to zero at the start of the movement (neutral position) than to visually determine the zero position by eye with the universal goniometer. In summary, the smartphone and universal goniometer mostly produced “Excellent” intra-rater reliability with small measurement errors and smaller MDC than Johnson et al. [1] and Furness et al. [2]. Therefore, one can repeatedly take measurements on separate days with the smartphone or the universal goniometer.

#### 4.4.2. Inter-Rater Reliability

Mostly “Excellent” inter-rater reliability with good agreement was achieved with the smartphone. “Good” to “Excellent” inter-rater reliability with moderate to good agreement was achieved with the universal goniometer. Therefore, the smartphone produced marginally better results than the bubble inclinometer. The results of the smartphone were also marginally better than the results achieved by Furness et al. [2] because smaller SEM and MDC was achieved. This indicates that the smartphone can be used to reproduce consistent results in an agreement between three raters over three days, indicating that the raters or clinicians can be interchangeable when taking repeated measurements with the smartphone over multiple days. However, one needs to interpret Furness et al. [2] with caution as they investigated the inter-rater reliability within one day.

#### 4.4.3. Validity

The smartphone achieved “Good” validity, but significant proportional bias was present. Furness et al. [2] also investigated the validity between a smartphone and universal goniometer and reported similar results. The proportional bias found in this current study indicates that at lower mean values, there was a tendency for the smartphone to overestimate the values and at higher mean values, there was a tendency for it to underestimate the values when compared with the universal goniometer. In addition, the large reference range indicates that there was large variability within the measurements. Therefore, the smartphone will not be clinically acceptable to measure the range of motion during the seated trunk rotation assessment interchangeably with the universal goniometer because of the lack of agreement between the two tools and large reference range. This large reference range indicates that there is low confidence for the smartphone to consistently agree with the universal goniometer. The variability within the measurements might have resulted from either the fatigue of the participants or represent the actual error of the tools. Therefore, the smartphone is not a valid tool to measure thoracic rotation during the seated trunk rotation assessment.

## 5. Conclusions

The Samsung s9+ smartphone is a reliable and valid tool that can be used to determine active and passive thoracic spine mobility of asymptomatic adults with lumbar-locked trunk rotation, standing relaxed curve and seated trunk rotation assessment. However, it did not produce clinically acceptable inter-rater reliability during the standing relaxed curve assessment and validity during the seated trunk rotation assessment. When selecting a tool or device for measuring thoracic spine mobility measurements, one should try to consistently use one device and avoid using different tools interchangeably. Sensors in smartphones are constantly improving and it would be expected that mobile applications would further improve with the sensors. However, future studies should focus on determining reliability and validity of these newly released sensors in smartphones and investigate symptomatic populations. The novelty of the findings within this study indicates that more research should be performed to determine the reliability of smartphones during lumbar-locked trunk rotation assessment. Therefore, it is recommended to do a small reliability and validity study on the smartphone before conducting a bigger research project with it. In addition, a standardized thoracic spine range of motion assessment protocol should be developed with the aim to eliminate variation within the results between raters or clinicians, to have clear guidelines to instruct them and to eliminate differences between research methodologies of studies. Overall, the smartphone is a reliable and valid tool to determine thoracic spine mobility, and it is easy to use and cost effective.

## Figures and Tables

**Figure 1 sensors-23-07622-f001:**
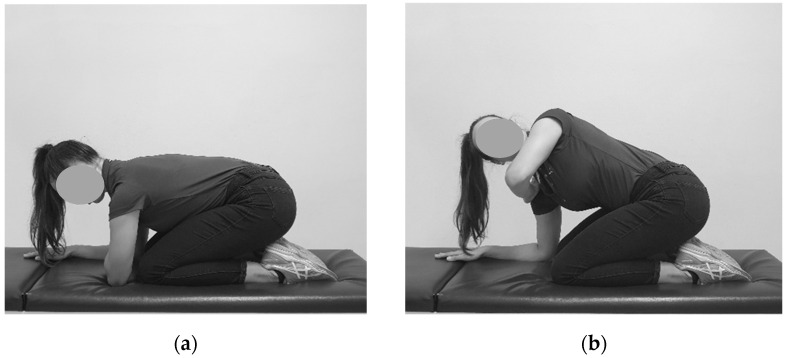
Lumbar-locked trunk rotation: (**a**) Starting position; (**b**) End position.

**Figure 2 sensors-23-07622-f002:**
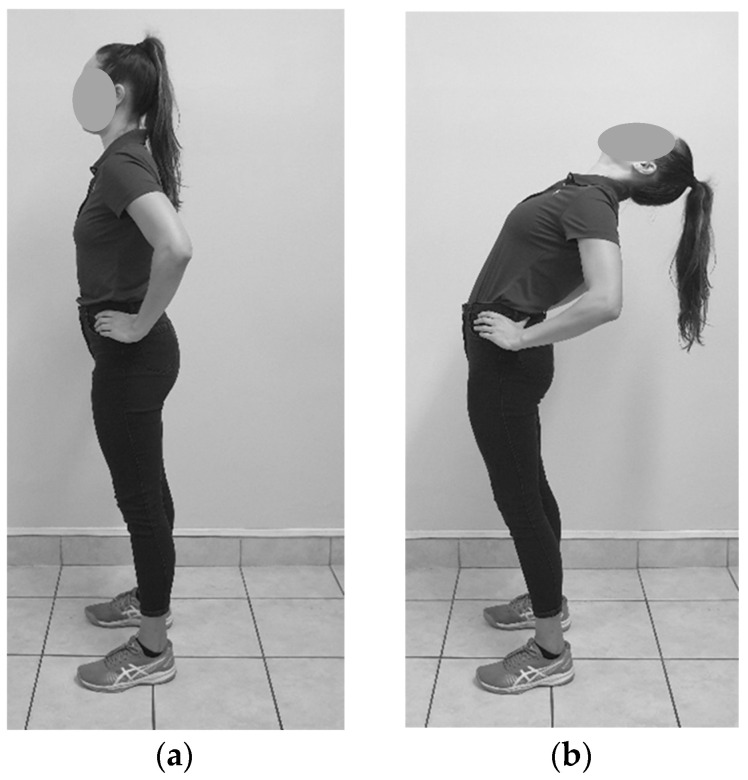
Standing full extension: (**a**) Starting position; (**b**) End position.

**Figure 3 sensors-23-07622-f003:**
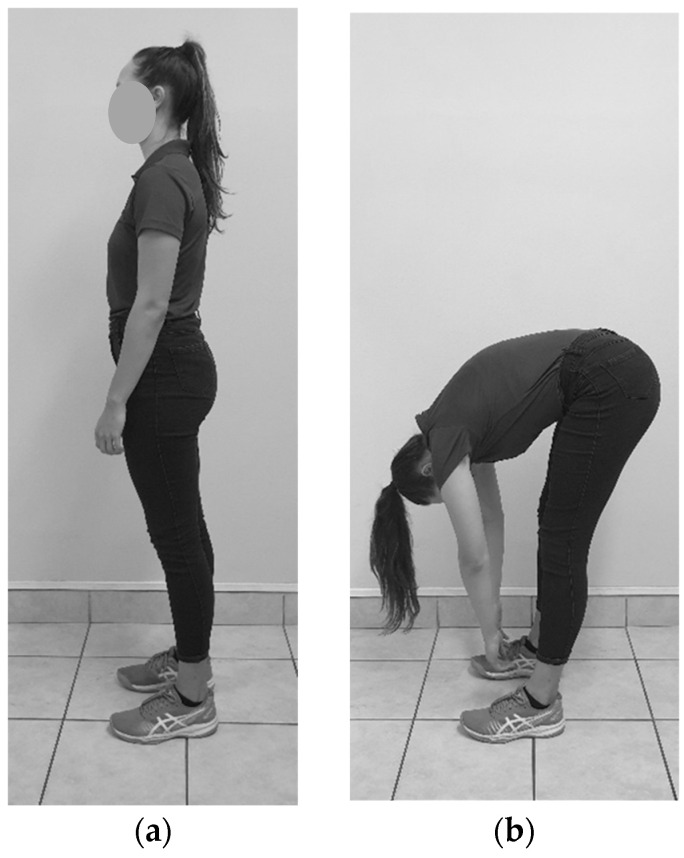
Standing full flexion: (**a**) Starting position; (**b**) End position.

**Figure 4 sensors-23-07622-f004:**
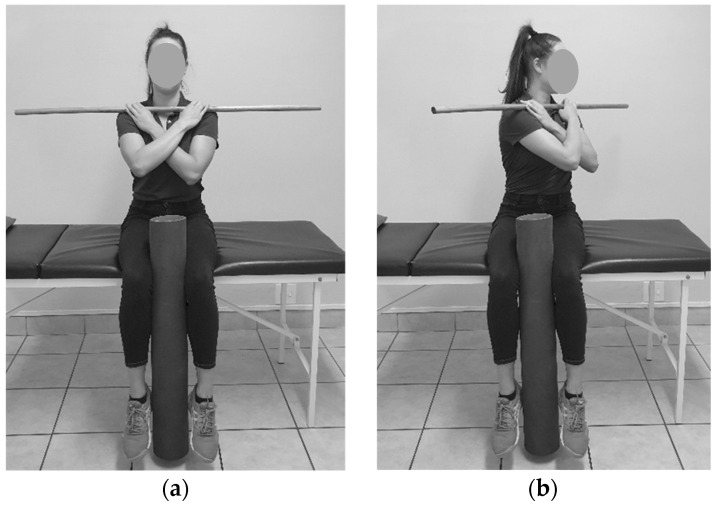
Seated trunk rotation: (**a**) Starting position; (**b**) End position.

**Figure 5 sensors-23-07622-f005:**
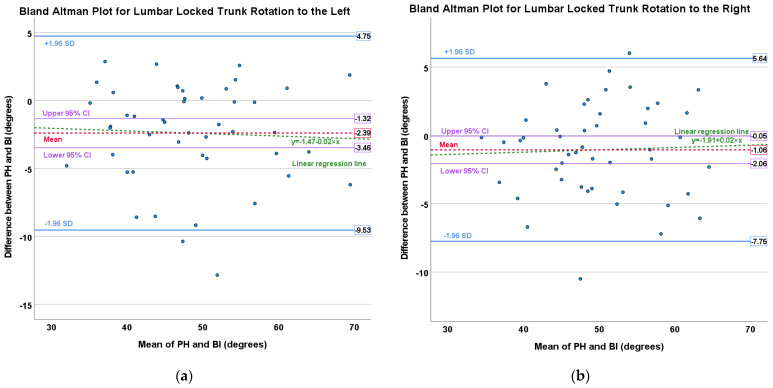
Bland-Altman plots indicating bias (95% CI) and proportional bias (linear regression line) between the preferred tool and smartphone with the following clinical assessments: (**a**) Lumbar-locked trunk rotation to the left; (**b**) Lumbar-locked trunk rotation to the right; (**c**) Standing relaxed curve; (**d**) Seated trunk rotation to the left; (**e**) Seated trunk rotation to the right. CI = confidence interval; SD = standard deviation; PH = smartphone; BI = bubble inclinometer; UG = universal goniometer.

**Table 1 sensors-23-07622-t001:** Summary of clinical thoracic spine assessments and clinical tools used.

Clinical Assessments	Clinical Tools	
	Preferred Tool	New Tool
Lumbar-locked trunk rotation	Bubble inclinometer	Smartphone
Standing full extension	Bubble inclinometer	
Standing full flexion	Bubble inclinometer	
Standing relaxed curve	Bubble inclinometer	Smartphone
Seated trunk rotation	Universal goniometer	Smartphone

Notes: Preferred tool = commonly used tool/s for clinical assessment/s. Smartphone = Samsung s9+ (iSetSquare mobile application).

**Table 2 sensors-23-07622-t002:** Intra-rater reliability of three raters.

Clinical Assessment	Rater (R)	Tool	ICC (95% CI)	SEM (°)	MDC (°)
LL Left	R1	Preferred tool	0.855 (0.679 to 0.943)	3.4	9.4
LL Left	R2	Preferred tool	0.866 (0.682 to 0.952)	4.1	11.3
LL Left	R3	Preferred tool	0.903 (0.774 to 0.964)	3.3	9.2
LL Left	R1	Smartphone	0.878 (0.727 to 0.952)	2.9	8.1
LL Left	R2	Smartphone	0.871 (0.698 to 0.953)	3.8	10.5
LL Left	R3	Smartphone	0.914 (0.798 to 0.969)	3.1	8.6
LL Right	R1	Preferred tool	0.776 (0.495 to 0.912)	4.3	11.9
LL Right	R2	Preferred tool	0.816 (0.566 to 0.933)	3.8	10.4
LL Right	R3	Preferred tool	0.761 (0.427 to 0.913)	4.4	12.1
LL Right	R1	Smartphone	0.739 (0.409 to 0.898)	4.2	11.6
LL Right	R2	Smartphone	0.889 (0.737 to 0.960)	2.9	8.1
LL Right	R3	Smartphone	0.890 (0.738 to 0.960)	2.8	7.7
SFE P1	R1	Preferred tool	0.955 (0.897 to 0.983)	4.0	11.0
SFE P1	R2	Preferred tool	0.955 (0.870 to 0.985)	3.1	8.5
SFE P1	R3	Preferred tool	0.889 (0.732 to 0.961)	5.2	14.3
SFE P2	R1	Preferred tool	0.973 (0.938 to 0.990)	3.0	8.2
SFE P2	R2	Preferred tool	0.950 (0.866 to 0.966)	3.7	10.4
SFE P2	R3	Preferred tool	0.888 (0.730 to 0.961)	4.9	13.7
SFF P1	R1	Preferred tool	0.969 (0.931 to 0.988)	2.8	7.8
SFF P1	R2	Preferred tool	0.880 (0.689 to 0.958)	4.8	13.3
SFF P1	R3	Preferred tool	0.854 (0.617 to 0.949)	5.7	15.8
SFF P2	R1	Preferred tool	0.981 (0.981 to 0.958)	1.9	5.2
SFF P2	R2	Preferred tool	0.934 (0.847 to 0.976)	3.5	9.8
SFF P2	R3	Preferred tool	0.917 (0.805 to 0.970)	3.6	10.1
SRC	R1	Preferred tool	0.929 (0.841 to 0.972)	1.7	4.7
SRC	R2	Preferred tool	0.691 (0.294 to 0.886)	4.3	11.9
SRC	R3	Preferred tool	0.822 (0.589 to 0.935)	2.8	7.6
SRC	R1	Smartphone	0.942 (0.872 to 0.977)	1.5	4.3
SRC	R2	Smartphone	0.624 (0.145 to 0.861)	4.0	11.1
SRC	R3	Smartphone	0.824 (0.545 to 0.938)	2.7	7.5
TR Left	R1	Smartphone	0.918 (0.816 to 0.968	2.6	7.2
TR Left	R2	Smartphone	0.908 (0.781 to 0.966)	2.6	7.3
TR Left	R3	Smartphone	0.933 (0.844 to 0.976)	2.1	6.0
TR Left	R1	Preferred tool	0.958 (0.906 to 0.983)	2.2	6.1
TR Left	R2	Preferred tool	0.940 (0.860 to 0.978)	3.0	8.4
TR Left	R3	Preferred tool	0.850 (0.652 to 0.945)	4.0	11.2
TR Right	R1	Smartphone	0.949 (0.886 to 0.980)	2.0	5.5
TR Right	R2	Smartphone	0.868 (0.560 to 0.957)	3.4	9.5
TR Right	R3	Smartphone	0.918 (0.802 to 0.970)	2.2	6.0
TR Right	R1	Preferred tool	0.920 (0.823 to 0.969)	3.2	9.0
TR Right	R2	Preferred tool	0.938 (0.852 to 0.978)	3.3	9.1
TR Right	R3	Preferred tool	0.701 (0.290 to 0.890)	4.5	12.6

Notes: ICC = Interclass correlation coefficient; CI = Confidence interval; SEM = Standard error of measurement; MDC = Minimal detectable change; LL = Lumbar-locked trunk rotation; SFE = Standing full extension; SFF = Standing full flexion; P1 = Position 1; P2 = Position 2; SRC = Standing relaxed curve; TR = Seated trunk rotation; R1 = Rater 1; R2 = Rater 2; R3 = Rater 3.

**Table 3 sensors-23-07622-t003:** Inter-rater reliability between three raters.

Clinical Assessment	Tool	ICC (95% CI)	SEM (°)	MDC (°)
LL Left	Preferred tool	0.936 (0.837 to 0.977)	2.3	6.4
LL Left	Smartphone	0.928 (0.793 to 0.976)	2.4	6.7
LL Right	Preferred tool	0.882 (0.600 to 0.962)	2.6	7.2
LL Right	Smartphone	0.860 (0.461 to 0.957)	2.8	7.7
SFE P1	Preferred tool	0.960 (0.904 to 0.986)	3.1	8.6
SFE P2	Preferred tool	0.968 (0.922 to 0.989)	2.8	7.8
SFF P1	Preferred tool	0.953 (0.899 to 0.983)	3.0	8.4
SFF P2	Preferred tool	0.957 (0.895 to 0.984)	2.6	7.3
SRC	Preferred tool	0.819 (0.555 to 0.935)	2.6	7.1
SRC	Smartphone	0.671 (0.080 to 0.890)	3.2	9.0
TR Left	Smartphone	0.942 (0.751 to 0.983)	1.9	5.4
TR Left	Preferred tool	0.958 (0.895 to 0.985)	2.1	5.9
TR Right	Smartphone	0.955 (0.896 to 0.984)	1.7	4.8
TR Right	Preferred tool	0.873 (0.519 to 0.961)	3.6	9.9

Notes: ICC = Interclass correlation coefficient; CI = Confidence interval; SEM = Standard error of measurement; MDC = Minimal detectable change; LL = Lumbar-locked trunk rotation; SFE = Standing full extension; SFF = Standing full flexion; SRC = Standing relaxed curve; P1 = Position 1; P2 = Position 2; TR = Seated trunk rotation.

**Table 4 sensors-23-07622-t004:** Descriptive data.

Clinical Assessment	Tool	Mean° (SE°)	(95% CI)	Residuals	
				Mean° (SE°)	(95% CI)
LL Left	Preferred tool	49.6 (1.3)	(46.9 to 52.3)	−2.4 (0.5)	(−3.5 to −1.3)
	Smartphone	47.2 (1.3)	(44.6 to 49.9)		
LL Right	Preferred tool	50.4 (1.1)	(48.2 to 52.7)	−1.1 (0.5)	(−2.1 to −0.1)
	Smartphone	49.4 (1.2)	(47.1 to 51.7)		
SRC	Preferred tool	33.3 (0.9)	(31.5 to 35.2)	−2.1 (0.5)	(−3.1 to −1.1)
	Smartphone	31.3 (1.0)	(29.3 to 33.3)		
TR Left	Preferred tool	57.2 (1.5)	(54.1 to 60.2)	−2.6 (1.0)	(−4.5 to −0.6)
	Smartphone	54.6 (1.2)	(52.2 to 57.0)		
TR Right	Preferred tool	57.2 (1.6)	(54.0 to 60.4)	−1.9 (0.9)	(−3.8 to 0.0)
	Smartphone	55.3 (1.2)	(53.0 to 57.7)		

Notes: SE = Standard error; CI = Confidence interval; LL = Lumbar-locked trunk rotation; SRC = Standing relaxed curve; TR = Seated trunk rotation.

**Table 5 sensors-23-07622-t005:** Validity between preferred tool and smartphone used to measure thoracic mobility.

Clinical Assessment	Tool	ICC (95% CI)
LL Left	Preferred tool vs. smartphone	0.942 (0.823 to 0.975)
LL Right	Preferred tool vs. smartphone	0.947 (0.902 to 0.971)
SRC	Preferred tool vs. smartphone	0.903 (0.744 to 0.956)
TR Left	Preferred tool vs. smartphone	0.836 (0.691 to 0.911)
TR Right	Preferred tool vs. smartphone	0.867 (0.759 to 0.927)

Notes: CI = Confidence interval; LL = Lumbar-locked trunk rotation; SRC = Standing relaxed curve; TR = Seated trunk rotation.

## Data Availability

The data presented in this study are available on request from the corresponding author. The data are not publicly available due to the institution’s Intellectual Property regulations.

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
