# Peer review of "Reliability and Validity of a Smartphone Device and Clinical Tools for Thoracic Spine Mobility Assessments"

_sensors, 2023, doi:10.3390/s23177622_

Round 1

Reviewer 1 Report

The manuscript under review focuses on assessing the reliability and validity of using a smartphone compared to common clinical tools for measuring thoracic spine mobility in asymptomatic adults. The study sets out two primary goals: (i) to determine the reliability of the smartphone and traditional clinical tools, and (ii) to evaluate the smartphone's validity against the established clinical tools in measuring thoracic spine mobility.

The authors present two hypotheses, the first being that the smartphone will provide similar categorical ICC scores for reliability as the widely used clinical tools. Additionally, they hypothesize that the smartphone will yield accurate results with minimal strain during usage.

Upon review, it is evident that the study is well-designed, with a clear structure and logical flow of information. The article is also commendable for its readability, making it accessible to a broader audience. The figures and tables provided in the manuscript are generally clear and aid in visualizing the findings. However, it is worth noting that Figure 5 requires improvement, as the font size used makes it difficult to read the text.

The authors have done an excellent job of citing up-to-date literature to support their research, demonstrating their thorough understanding of the topic. This contributes to the manuscript's credibility and relevance within the field. In conclusion, I recommend the publication of this manuscript in its current form, as it provides valuable insights into the potential use of smartphones as a reliable and valid tool for measuring thoracic spine mobility in asymptomatic adults. Addressing the font size issue in Figure 5 will further enhance the overall quality of the paper. This study has the potential to be of great interest to researchers and healthcare professionals alike, offering a novel approach to assessing thoracic spine mobility.

Author Response

We would like to thank reviewer 1 for their constructive feedback and suggestions regarding figure 5. We have made changes to figure 5 to make it visually appealing.

Reviewer 2 Report

this is a very well designed and well described study, thank you very much.  

Author Response

The authors would like to thank reviewer 2 for their positive feedback.

Reviewer 3 Report

Reliability and validity of a smartphone device and clinical tools for thoracic spine mobility assessments

The authors conducted a study to analyze the intra- and inter-observer reliability of two measurement systems. One commonly used by clinicians and the other by means of a smartphone application. A reliability and validity study of a measurement instrument is always welcome and the chosen body area (thoracic spine) shows a great clinical relevance to measure and evaluate this area accurately. The use of smartphones as biomechanical assessment tools is becoming more and more widely used and therefore studies are needed to evaluate the reliability of these tools.

I have some questions or suggestions regarding your manuscript:

Introduction

In addition, we hypothesized that the smart phone device will produce 64 valid results with minimal bias.

This word is separated

Methodology

Study design

This study follows a quantitative experimental study design. What type of study are the authors referring to? This is not an experimental study as there is no intervention.

Participants

Physical Activity Readiness Questionnaire (PAR-Q): the authors should refer to this tool.

Analysis

Validity: I think it would be more appropriate to assess validity with Pearson's product-moment correlation coefficient (r).

Regarding the measurements, I have a concern:  why were full flexion and extension in standing position selected. The thoracic spine is a difficult region to evaluate since the isolated movement of this region is complex to isolate. In relation to this point, I believe that these two measurements are not the most appropriate for assessing the thoracic region, since both the lumbar spine and the pelvis contribute to these movements. And why wasn't full flexion and extension assessed with the mobile?

The results should be better displayed. The figures should be self-explanatory and help the reader to interpret them.

In the discussion, although the authors comment on them at several points, the limitations of this study should be made clear. On this point I observe some inconsistency since in the methodology the authors comment that the evaluators are trained and experienced persons, however, to justify the low intraobserver reliability in Lumbar-locked trunk rotation, the authors comment that the evaluators did not have enough experience.

 Minor editing of English language required

Author Response

We would also like to thank reviewer 3 for your questions and suggestions. We have addressed the suggestions within the article and below:

Analysis:

Thank you for your suggestion regarding the use of Peason’s product-moment correlation coefficient (r) for determining the validity. We chose the intra-class correlation coefficient (ICC) because it looks at the agreement and it considers the differences in the means measured (Liu, Tang, Chen, Lu, Feng & Tu, 2016). The use of the ICC also gave us the ability to compare our results to recently published studies.

Liu, J., Tang, W., Chen, G., Lu, Y., Feng, C. & Tu, X.M. 2016. Correlation and agreement: overview and clarification of competing concepts and measures. Shanghai Archives of Psychiatry. 28(2):115–120. DOI: 10.11919/j.issn.1002-0829.216045.

Regarding measurements:

Thank you for raising your concern regarding the standing flexion and extension measurements. The reason for doing these assessments standing was to be done according to traditional way of measuring the spinal range of motion with the bubble inclinometer. This allowed us to compare our results to previous research.

We agree that it is difficult to isolate some of the movements of the thoracic spine but the measurement sites were limited to the thoracic region. Therefore, eliminating the contribution from the pelvis within the measurements. The novelty of this study highlights the need to construct alternative assessments specifically for the thoracic spine but these new assessments would need to be standardized and be compared to the traditional spine assessments.

The reason for not measuring the full flexion and extension with the smartphone is because you require two smartphones to measure the two measurement sites simultaneously. The primary investigator did not have access to two identical smartphones for this assessment.

Discussion:

Thank you for the comment regarding the limitation of “not enough experience”. We have addressed this by editing the language to make it clear that it is not the clinical inexperience that led to the low intra-rater reliability but rather the lack of experience of conducting the specific assessments included within the study.
